# Neurodynamic Functions and Their Correlations with Postural Parameters in Adolescents with Idiopathic Scoliosis

**DOI:** 10.3390/jcm11041115

**Published:** 2022-02-19

**Authors:** Agnieszka Stępień, Beata Pałdyna

**Affiliations:** 1Department of Rehabilitation, Józef Piłsudski University of Physical Education, 00-968 Warsaw, Poland; 2ORTHOS Functional Rehabilitation Centre, 02-793 Warsaw, Poland; beata.paldyna@gmail.com

**Keywords:** idiopathic scoliosis, neurodynamic functions, assessment, pain, treatment

## Abstract

Knowledge about neurodynamic functions of the nervous system (NS) in patients with idiopathic scoliosis (IS) is limited. This study aimed to assess the mechanosensitivity of the NS structures (MNS) in adolescents with IS. The study included 69 adolescents with IS and 57 healthy peers aged 10–15 years. The Upper Limb Neurodynamic Test 1 (ULNT1), straight leg raise (SLR) test, and slump test (SLUMP) were used to assess MNS. The spinal curvatures in the sagittal plane and selected ranges of motion were measured. The data were analysed using the Mann–Whitney U test and Spearman’s rank correlation. Increased MNS assessed by ULNT1 and SLUMP tests was observed in participants with IS. Values of the neurodynamic tests correlated significantly with the sagittal profile of the spine and the mobility of the spine and lower limbs in both groups. In conclusion, increased MNS occurs in adolescents with IS. Therefore, the examination of adolescents with IS should include an assessment of MNS with the neurodynamic tests. Future studies should investigate this issue to better understand the mechanisms that coexist with IS.

## 1. Introduction

Various co-existent abnormalities have been described in patients with idiopathic scoliosis (IS), but knowledge about the neurodynamic functions of the nervous system (NS) is still limited [1,2].

Neurodynamics is the science of the correlations between the mechanics and physiology of the nervous system, as well as between assessment and therapy [3,4,5]. This field of medicine has been developing intensively for several decades.

The literature indicates that tensile loading is necessary for mechanical adaptation and the normal functioning of neurons and nerves [3,4]. Various aspects of adverse neural mechanics have been presented in the past. It is known that mechanical disorders, such as disturbed tensile strength, nerve gliding disturbance (the limited movement of the neural structures relative to other tissues) or compression (pressure-induced deformation) of the nerve, may decrease neural tissue extensibility and cause irritation. In this case, these structures become sensitive to tensile forces [3]. Some authors of the neurodynamic concept strongly emphasised that many neural problems have their causes in the musculoskeletal system. They suggested that the structures of NS follow the body’s movements and are mechanically loaded during daily activities [3].

Mechanosensitivity of the NS structures (MNS) can be detected with the neurodynamic tests. These tests are designed to apply tensile forces to the tissues of NS. Elongation is applied to the neural tissues by increasing the distance between the ends of the nerve tract [3]. Impaired MNS may be manifested by pain, reduced range of motion or sensory disturbances. The neurodynamic tests were used mainly to examine the patients with various impairments in the musculoskeletal system and evaluate the effects of the therapy. 

Tensioning techniques were the first neurodynamic techniques applied to treat patients with neuropathies. Later, sliding techniques were developed [4]. The positive effect of neural mobilisation on neuromusculoskeletal conditions has been observed, especially its benefits for back and neck pain [6,7]. Current studies analysing tensioning techniques revealed neuroimmune, neurophysiological, and neurochemical effects in the structures of the nervous system [4].

Previous studies have demonstrated various tests such as (1) the upper limb neurodynamic test 1 (ULNT1), which assesses length and mobility of the peripheral NS in an upper limb [8,9], (2) the straight leg raise (SLR) test with ankle dorsiflexion, which is often used in patients with back and lower limb pain [10,11,12,13], or (3) the slump (SLUMP) test, which assess the correlations between the patient’s symptoms and movement limitations in the pain-sensitive structures located in the spinal canal and intervertebral foramen [10,12]. The studies have proven the reliability of ULNT1 [14,15,16], SLR [13,17,18], and SLUMP tests [17,18]. Moreover, the ULNT1, SLR, and SLUMP tests have been used mainly to assess back and leg pain among adults [10,12]. However, the utility of the tests in individuals without symptoms has also been investigated [11,16,19]. 

A few studies revealed that a limited range of motion in the SLR test might also be applied during the period of childrens’ growth [9]. The SLR test was also used in a study of children with cerebral palsy [20]. In addition, the modified long sitting slump test was applied to examine children with headaches compared to healthy controls [21]. However, no study has examined neurodynamic functions in adolescents with IS. There have been no analyses of correlations between neural tissue extensibility, alignment of spinal curvatures in the sagittal plane, angle of trunk rotation (ATR), or ranges of motion. Given that a three-plane change in the position of the spine axis alters the vectors of forces acting on the body, spinal misalignment in individuals with IS may theoretically affect MNS. 

Current literature presents different opinions regarding back pain in adolescents with scoliosis. Some researchers have reported back pain in this population [22,23,24,25], but others have argued that pain is not the primary problem related to IS [26]. In establishing the goals of IS treatment, experts of the International Society on Scoliosis Orthopaedic and Rehabilitation Treatment (SOSORT) indicated aesthetics, quality of life, and disability as the most important. However, pain prevention and reduction were included as equally important goals. The SOSORT guidelines have not included recommendations for assessing neurodynamic functions in individuals with IS [2,27].

Given the lack of knowledge regarding the neurodynamic of the NS in individuals with spine deformity, this study aimed to assess MNS, expressed by ranges of motion in neurodynamic tests, in adolescents with IS. Moreover, the correlations between MNS and selected postural parameters were analysed. Finally, the results were compared with those obtained in a group of healthy peers.

## 2. Materials and Methods

The study was conducted after being approved by the Senate Bioethical Commission at the Józef Pilsudski University of Physical Education in Warsaw (SKE 01-30/2020). The participants were informed about the study aims during consultations in the physiotherapy centre specialising in postural disorders prevention and spinal deformities treatment. 

Participants self-reported directly to physiotherapists without referrals from other specialists. In addition, legal guardians of children gave written consent to participate in the study.

### 2.1. Participants

The criteria for inclusion in the study group were as follows: girls and boys aged 10–15 years, single-curve (a left-sided lumbar/thoracolumbar curve) or double-curve (a right-sided thoracic curve and a left-sided lumbar/thoracolumbar curve) IS confirmed by radiological examination with Cobb angle value of more than 10°, absence of central nervous system disorders, chronic respiratory system diseases, metabolic or oncological diseases, injuries, and fractures within the previous 6 months. The control group included girls and boys aged 10–15 years without systemic diseases and with an angle of ATR of 0–5°, as measured by a Bunnell scoliometer. 

### 2.2. Measures

The study was carried out during individual physiotherapeutic consultations in the rehabilitation centre. The participants were examined by two physiotherapists with over 25 and 15 years of experience treating patients with IS. Both physiotherapists completed manual therapy training, including examination with neurodynamic tests. They met three times before the beginning of the study to unify the study methodology and manner of performing the tests. The physiotherapists were not blinded to the existence of scoliosis when examining the participants, as symptoms of scoliosis are usually visible.

At the beginning of the study, basic information regarding IS, the type of treatment (physiotherapy, brace), the occurrence of pain in the last 3 months, and pain localisation was collected. The question of pain prevalence was general. The type of pain and its occurrence and severity were not assessed. 

The study included the following measurements: (1) body height and mass, (2) the ranges of motion in ULNT1, SLR, and SLUMP on the left and right side of the body, (3) ATR, (4) spinal curvatures in the sagittal plane (cervicothoracic junction—C, thoracic upper—T1, thoracolumbar junction—T2, lumbosacral junction—LS), (5) ranges of cervical rotation (CR), (6) hip joint extension with flexion of the knee (HE), (7) hip joint flexion with the extension of the knee joint (HF), (8) and rotation mobility of the lumbo-pelvic-hip complex with trunk-pelvis-hip angle (TPHA) on both sides of the body, (9) the fingertip-to-floor (FTF) test, and (10) generalised joint hypermobility (JHM) using the Beighton scale.

The abbreviations of the tests used in this study are presented in Figure 1. In addition, the test methodology is described below.

The ULNT1 test [14,15,16] was performed with a patient lying on a table with an upper limb abducted by 90° and the head in a neutral position. The physiotherapist stabilised the arm and elbow of the patient with one upper limb, while the opposite upper limb performed wrist dorsiflexion, finger extension, forearm supination, shoulder joint external rotation, and elbow extension until the first feeling of tension. Verification of the affected structure was performed by moving the head to the tested side. Decreased symptoms meant disturbed NS slides. The range of motion was measured with a goniometer by placing the rotational axis along the axis of the elbow joint (Figure 2A,B). Lower values of the ULNT1 ranges meant greater neural tissue extensibility.

The SLR test [11,13,19] was performed in a supine position. The physiotherapist initially arranged the patient’s ankle joint in a neutral position, blocked the knee joint in extension, and then passively raised the lower limb until the first sign of tension. The patient’s feelings were verified by plantar flexion in the ankle joint. Decreased symptoms meant disturbed peripheral NS slides. The measurement was performed with a Rippstein plurimeter zeroed horizontally and placed at the lower leg distally from the tibial tuberosity (Figure 2C,D). Higher values of SLR ranges meant greater extensibility of neural structures.

SLUMP test [17,18] was performed on a patient in a sitting position, with the neck and trunk bent forward and the sacral bone in a vertical position. The patient joined the upper limbs behind the back. A physiotherapist placed one hand on the back of the patient’s neck and extended the patient’s knee joint with the other hand until the first feeling of tension, at the same time maintaining the ankle joint in a neutral position. The measurement was made with a plurimeter placed at the lower leg below the tibial tuberosity. Verification of the affected structure was made based on the extended cervical segment of the spine (moving the chin away from the neck) (Figure 2E,F). Decreased symptoms meant disturbed NS slides. Lower values of the SLUMP ranges indicate greater neural tissue extensibility. Measurements with a plurimeter during SLR and SLUMP tests have been used in previous studies [12].

The ATR measurement in the thoracic and lumbar segments was performed with a scoliometer in a standing position, with the trunk bent forward. The reliability of this measurement has been assessed in the past. This test is widely used in screening for scoliosis and assessing the effects of treatment in patients with scoliosis [28,29].

The position of the spine in the sagittal plane was assessed with the Rippstein plurimeter zeroed vertically and placed at the cervical-thoracic junction (C), in the upper part of the thoracic spine (T1—spine segment Th1–Th3), at the thoracic–lumbar (T2—spine segment Th11–L1) and lumbosacral junction (LS—spine segment L5–S1). Moreover, the position of the sternum was measured by placing a plurimeter on the upper part of the sternum. In previous studies based on a similar methodology, the Rippstein plurimeter was used to assess the curvatures of the spine in the sagittal plane in children with scoliosis and healthy controls [30,31].

The ranges of hip flexion with knee extension and extension in the hip joint with knee flexion were measured in a supine position according to a widely applied methodology [32]. A plurimeter zeroed parallel to the floor was placed on the thigh above the patella when the hip joint was extended. In the past, the measurement of extension in hip joints made with the plurimeter proved to be a reliable test in children with neuromuscular disease [33]. Therefore, when measuring flexion in the hip joint with extension in the knee joint, the plurimeter was placed at the lower leg below the tibial tuberosity, similarly to the SLR and SLUMP tests. 

The mobility of the lumbo-pelvic-hip complex was measured using the TPHA test using a plurimeter zeroed parallel to the floor. Values below the horizontal (greater ranges of movement) were designated as “+” in this study, and values above the level (movement restriction) were designated as “–”. Previous studies have confirmed the good reliability of the TPHA test, which was used in adolescents with IS and their healthy peers [34,35]. 

Also, the FTF test has been used in previous studies in people with back pain. The good validity of this test has been demonstrated [13,36].

### 2.3. Statistical Analysis

Statistical analyses were performed using STATISTICA version 13. Normal distribution was assessed with the Shapiro–Wilk test. Due to the lack of a normal distribution, the Mann–Whitney U test was used to determine any differences between the tested parameters in both groups. Spearman’s rank correlation coefficient was used to analyse the correlations between the parameters. Correlations were interpreted as: <0.3, —negligible correlation; 0.3–0.5, low correlation; 0.5–0.7, moderate correlation; 0.7–0.9, high correlation; and >0.9, very high correlation [37]. Quantitative variables were described as median ± quarter deviation; however, means ± SD were presented as additional information. The qualitative variables were analysed using the chi-square test. We adopted *p* = 0.05 as the level of significance.

## 3. Results

The study included 126 individuals: 69 patients with IS and 57 controls without scoliosis (C). The groups did not differ significantly in terms of age, body mass, or body height. In both groups, there were more girls than boys. There were definitely more adolescents with double-curve scoliosis in the IS group than with single-curve scoliosis. Within this group, the Cobb angle range was 11° to 69°. In total, 25 participants (36.3%) had low scoliosis at 11–20°, 41 were diagnosed with scoliosis at 21–50° (59.4%), and 3 participants (4.3%) had a Cobb angle greater than 50°. Most of the IS participants confirmed their participation in physiotherapy sessions. The pain was more common in adolescents with spinal deformities than in the control group. In both groups, the pain was most often located in the thoracic and lumbar segments of the spine. Detailed information on all of the participants is included in Table 1.

### 3.1. Neurodynamic Tests, Postural Parameters, and Range of Motion

Significantly lower ranges of ULNT1 and SLUMP tests (lower extensibility) were observed on both sides of the body in participants with scoliosis compared to the control group. The SLR test values did not differ significantly (Figure 3, Table 2). Increased ULNT1left values (5° or more) were found in 36 adolescents with scoliosis (52.2%) and 12 (17.4%) healthy participants. Increased ULNT1right values occurred in 36 participants with scoliosis (52.2%) and 14 (20.3%) controls. The values of SLRleft and SLRright less than 60° (lower extensibility) were achieved by 36 participants with IS (52.2%) and 26 controls (45.6%). A group of 49 adolescents with IS (71.0%) and 23 participants without scoliosis (40.4%) achieved a SLUMPleft of more than 20° (higher neural tension). A SLUMPright greater than 20° was found in 53 participants with IS (76.8%) and 22 healthy (38.6%) participants.

During the analysis, we checked how many participants had unequal ranges of neurodynamic tests on the left and right sides of the body, assuming a 5° difference as asymmetry. In 31 (44.9%) individuals with scoliosis, there were differences between the ULNT1 ranges on the left and right sides, equal to or larger than 5°. In the control group, asymmetry was noted in only 10 (17.5%) out of 57 adolescents. The SLR test revealed a minimum 5° difference in 23 (33.3%) participants with scoliosis and 8 controls (14.0%). The SLUMP measurements were not equal (minimum 5°) in 23 (33.3%) adolescents with spine deformity and in 8 (14.0%) participants without scoliosis. The differences between the mean values of the neurodynamic tests (ULNT1, SLR, and SLUMP) on both sides of the body were not significant in the IS patients and the control group. Detailed analysis of left/right asymmetry in the neurodynamic tests showed that limitation of the range of motion occurs with a similar frequency of 40–60% on both the left and right sides of the body in participants with IS (limitations: ULNT1left 54.8%, ULNT1right 45.2%, SLRleft 43.5%, SLRright 56.5%, SLUMPleft 47.8%, SLUMPright 52.5%).

The values of ATR in the thoracic and lumbar segments of the spine and Beighton score values were significantly higher in individuals with scoliosis. Measurements with the plurimeter revealed decreased values of T1 and T2 in the IS group compared to the control group, which indicates decreased kyphosis in patients with scoliosis. No significant differences were noted between the values of the measurements at the sternum, cervical-thoracic (C), and lumbosacral junction (LS). Further, the ranges of movement in the cervical spine, hip joints, and FTF test did not differ significantly (Table 2).

The analysis carried out in the subgroups of girls with scoliosis (*n* = 57) and without scoliosis (*n* = 42) revealed significant differences between the values of ULNT1left, ULNT1right, SLUMPleft and SLUMPright tests. The values of the SLR did not differ significantly. Girls with scoliosis demonstrated lower values of upper thoracic segment T1 inclination and higher values of ATRT, ATRL, and Beighton points (IS 3.73 ± 2.11; C 2.54 ± 2.47; *p* = 0.005) (Figure 4). No differences between the ranges of motion were found. Only a trend towards a lower range of TPHAright motion was observed in girls with scoliosis.

### 3.2. Correlations between Neurodynamic Tests and Postural Parameters

There was no significant relationship between the neurodynamic tests’ values and the Cobb angle values in adolescents with IS. However, participants with deformation over 30° showed significantly lower ranges of motion in the SLUMPleft test (*p* = 0.037). The IS and control groups revealed medium and low correlations between the ULNT1, SLR, and SLUMP tests. The increase in nerve tension in one test was accompanied by increased tension in the others. With an increase in the ULNT1 values, the ranges of SLR decreased, while higher SLUMP values accompanied an increase in the ULNT1 values (Table 3).

In both girls with scoliosis and their healthy counterparts, a significantly low correlation was observed between the position of the C and T1 segments in the sagittal plane and the ULNT1 and SLR tests values. The increased angular inclination of the spine from the axis was accompanied by higher MNS. In the IS group, a low positive correlation was observed between ULNT1 and the position of the lumbosacral segment (LS). Higher values of pelvic inclination were associated with higher values of ULNT1 test (lower extensibility). A weak correlation between the pelvic position (LS) and SLR values was found in the control group. The increase in ATR was correlated with the increased range in the ULNT1 test in this group (Table 4).

The analysis showed several relationships between neurodynamic tests and range of motion. In both groups, a significant correlation was found between the ULNT1, SLR, and SLUMP tests and the range of flexion in the hip joint with the knee extended (HF). The grater HF ranges were accompanied by greater neural extensibility. The FLF test showed a low correlation with SLR in the IS group. A moderate correlation was found between FLF, SLR, and SLUMP in the control group. No significant relationship was found between JHM and neurodynamic tests in either group (Table 5).

In the IS group, the ranges of TPHAright were correlated with ULNT1 and SLR. Higher ULNT1 values (lower extensibility) were associated with greater ranges of motion in both TPHAright (better mobility), and SLR was correlated with lower values of the mobility of the lumbo-pelvic-hip complex (limited mobility) (Table 5).

### 3.3. Neurodynamic Tests and Pain

The values of neurodynamic tests obtained by adolescents with and without pain in both groups were compared to confirm or exclude the effect of pain on the measurements. In addition, the values were analysed in the group of participants with IS, considering a subgroup of adolescents with and without pain.

Significant differences were found between adolescents without pain with scoliosis (*n* = 43) and those without scoliosis (*n* = 50). In participants with IS there were reduced ranges of motion in the ULNT1left (*p* < 0.001), ULNT1right (*p* < 0.001), SLUMPleft (*p* = 0.002) and SLUMPright (*p* = 0.001) tests. Due to the insufficient number of adolescents in the control group with pain (*n* = 7), the differences between the groups of participants with pain are not discussed.

The analysis of adolescents with IS revealed that patients with pain (*n* = 26) showed increased mechanosensitivity in the SLRright (*p* = 0.04) and SLUMPright (*p* = 0.006) tests compared to participants without pain (*n* = 43).

## 4. Discussion

The multivariant aetiology of IS, various health problems, and long-lasting treatment makes it necessary to conduct scientific studies aimed at a detailed description of this disease and the prevention of its symptoms in the musculoskeletal system. Unfortunately, current knowledge about the neurodynamic of the NS in patients with IS is limited. Therefore, this study aimed to evaluate MNS in adolescents with IS in comparison to a control group, which is an issue that has not previously been addressed in the literature.

The study included girls and boys with IS and asymptomatic adolescents aged 10–15 years. According to the World Health Organization, adolescents are defined as the period between childhood and adulthood, from ages 10 to 19 [38]. However, it is known that in this age, a rapid growth spurt and other puberty-related changes are observed [39,40]. Intensive changes related to maturation may affect posture and range of motion, and promote the development of scoliosis, making clinical reasoning difficult. The varied ages of the participants qualified for our study can therefore be considered factors that may affect the values of neurodynamic tests, postural parameters, and range of motion. It is worth conducting similar studies in different age categories among adolescents in the future.

For this study, three neurodynamic tests whose reliability had been previously confirmed in different groups of participants were selected: the ULNT1 [14,15,16], SLR [13,19,41], and SLUMP test [17,18]. In addition, we also applied other tests that have been used to assess patients with various dysfunctions, including scoliosis.

The results showed that MNS in adolescents with IS assessed using ULNT1 and SLUMP was larger than in the control group. No differences were noted in the ranges of SLR between the groups. These results indicate that adolescents with IS in puberty experience increased nerve sensitivity, but only in certain parts of NS. The upper part structures of the NS appeared to be more sensitive to stretching. The ULNT test used in this study was applied, inter alia, to assess the tension of the median nerve, that is, one of the nerves from the subclavicular part of the brachial plexus. During the SLUMP test, the structures of the spinal canal and intervertebral foramen in all segments of the spine were stretched. The SLR test assesses the extensibility of the nervous structures in the lower peripheral NS [3]. The study was conducted with a group of participants with various values of the Cobb angle. There was no correlation between the Cobb angle values and the parameters tested. However, participants with deformation over 30° showed a significantly lower range of motion in the SLUMPleft test than participants with less severe scoliosis, which may indicate some relationship between spine deformation and MNS.

Statistical analysis of painless participants showed increased mechanosensitivity in ULNT1 and SLUMP tests in adolescents with IS. At the same time, the increased range of motion in the SLRright and SLUMPright tests was observed in participants with IS and pain compared to their peers with IS, but without pain. These results indicate that both spinal deformities and the presence of pain may influence the MNS in adolescents.

Unfortunately, these results cannot be correlated with those of other studies since no similar studies have been conducted in adolescents with scoliosis to date. Further, no norms for neurodynamic functions have been defined in healthy children and youth groups. A few years ago, the reliability of the SLR test was assessed in a group of children with cerebral palsy [20]. Other studies have shown a higher intensity of the sensory response rate in the examination using the long sitting slump (LSS) test in children aged 6–12 years with a migraine or cervicogenic headaches than their healthy peers [21].

In our study, the analysis of the frequency of occurrence of differences between the ranges of motion on the left and right sides (equal to or larger than 5°) revealed that regardless of the type of neurodynamic test (ULNT1, SLR, SLUMP), such differences occurred more often in the IS group. These results suggest that scoliosis can lead to extensibility differences between the sides of the body.

In a study by Stalioraitis et al. [42] conducted on a group of asymptomatic individuals, no significant differences were noted in the range of motion in the ULNT1 test on the left and right sides of the body. Interestingly, our results revealed that differences between the mean values of neurodynamic tests (ULNT1, SLR, and SLUMP) on both sides of the body were not significant in either group, even though the frequency of asymmetry of measurements was higher in the IS group. It has also been observed that reduced range of motion in neurodynamic tests occurs in patients with IS at a similar frequency of 40–60% on both sides of the body. However, due to many variables such as Cobb angle, number and location of curvatures, the dominance of one of the curves in double-curve scoliosis, value of ATR, alignment in the sagittal plane, pain incidence, gender, hypermobility, the relationship between the left/right asymmetry and direction of the curvatures was not analysed in detail in this study. This phenomenon requires further analysis in a larger and more homogeneous group of adolescents with IS.

The limited range of motion in one neurodynamic test was accompanied by a significant decrease in other neurodynamic tests in both groups. This observation is valuable from a practical point of view. The finding of increased neural tension in one part of the nervous system may require therapy in other parts of the body, which is important because of the risk of irritation of the nerves and pain provocation [3,4,5].

The analysis of measurements in our study revealed decreased values of thoracic kyphosis in the IS group. This finding supports findings obtained by other authors who described changes in spinal curvature in the sagittal plane and suggested their influence on the development of scoliosis [43,44]. Numerous correlations were noted between the neurodynamic tests and the selected postural parameters. It appeared that the position of the C, T1, and LS segments in the sagittal plane was a significant factor affecting the values of the ULNT1 and SLR tests. This correlation could indicate that changes in the spine curvatures in the sagittal plane observed in patients with scoliosis affect MNS changes [43,44].

Our results also confirmed the theory that adolescents with IS experience a significantly higher JHM prevalence than their healthy peers [45]. In the study by Czaprowski et al. [45], JHM was observed in 51.4% of participants with IS and 19% in the control group, whereas in the present study, hypermobility was found in 33.3% of participants with IS and 14% of controls. No significant correlation was found between JHM and neurodynamic measurements in our study.

No differences were observed between the range of motion values in participants with IS and their control counterparts. The analysis showed that there are biomechanical connections between neurodynamic tests and range of motion. These correlations confirm that NS viscoelasticity is not only associated with the static position of the body, as mentioned above (sagittal alignment), but also with the mobility of the body. The HF test showed a significant relationship with all neurodynamic tests. The larger HF ranges were accompanied by lower NS tension in both groups. These results indicate that the HF test could be used to diagnose children in everyday clinical practice. However, this thesis requires confirmation in a larger group of children and adolescents with IS and those who are asymptomatic.

A comparison of the TPHA values achieved in both subgroups of girls showed that participants with IS tended to limit the range of rotational movement of the lumbo-pelvic-hip complex to the right (TPHAright). In a previous study by Stępień et al. [34] significant difference was observed between the TPHAright values in girls with IS and healthy peers. Further, a positive effect of rotational mobilisation on TPHA and ATR values was demonstrated in girls with double-curve scoliosis [35]. Earlier studies have also shown limitations in the range of rotation of the trunk and pelvis in adolescent girls with IS [46]. Rotational movement disorder was also recognised by Burwell et al. [47] as one of the factors in the development of scoliosis.

This study demonstrated a relationship between the TPHAright values and neurodynamic measurements in the IS group. As the TPHAright range increased, the mechanosensitivity increased in the ULNT1 and SLR tests. This relationship indicates a biomechanical relationship between neural tissues extensibility and rotational movements of the spine in individuals with scoliosis. Perhaps limiting the range of rotational movements of the spine leads to compensatory impairments of the NS. This theory, however, requires confirmation in future projects.

Increased NS irritation is often diagnosed in individuals experiencing pain. The literature reveals that pain occurs in adolescents with IS [22,23,24,25,26]. Prevention and reduction of pain are among the most important goals of conservative treatment for individuals with IS, although not the most important [2,26,27]. It is assumed that if Cobb’s angle is above 30°, the risk of pain, health problems in adult life, and limitations in an everyday functioning increase [2].

Our results indicate that pain was observed significantly more often in adolescents with IS (37.7%) than in healthy controls (12.3%). A localised pain in different parts of the body occurred more often in participants with curvature angles higher than 30° (50.0%) than in patients with scoliosis below 30° (31.1%) or in healthy individuals (14%). This could indicate a relationship between three-dimensional spinal deformity and back pain, but in most studies, the pain has not shown a strong correlation with the Cobb angle [26]. The prevalence of back pain in our participants (29.0%) was less than in the population assessed by other authors. Teles et al. [25] showed that 90% of adolescents with IS reported back pain, mostly mild in intensity (37.5%). Sato et al. [22] conducted an epidemiological study among students aged 9–15 years. The authors showed a 58.8% prevalence of back pain in scoliotic patients, compared with 33% in non-scoliotic students. In a study by Théroux et al. [23], 47.3% of patients with IS aged 10–17 years had back pain, with the most severe pain in the lumbar region.

In studies by Wong et al. [24], depending on the analysed period (12 months, 30 days, 7 days), the prevalence of thoracic pain ranged from 6% (within 12 months) to 14% (within 7 days), whereas that in the lumbar region ranged from 6% to 29%. These results are similar to ours, where 17.4% of the IS adolescents complained the thoracic segment pain, and 21.7% had low back pain.

The frequency of occurrence of back pain in our project among children without scoliosis seems to be lower than in other studies, in which over 20% of healthy adolescents reported low back pain [48,49]. For example, studies conducted in Poland have shown that 12.2–61.5% of children aged 10–13 years and 14–65.7% aged 14–16 years report back pain, depending on the frequency of occurrence, location, intensity, and situations [50]. However, in our opinion, our results cannot be compared with others since our project did not consider the location and severity of pain, nor did we use reliable pain assessment methods.

SOSORT recommends that physiotherapeutic scoliosis-specific exercises (PSSEs) should be adapted to individual needs, including pain prevention or reduction in IS patients [2]. In our opinion, because pain occurred more often in adolescents with IS and the differences between neurodynamic tests in IS and healthy participants, impaired MNS should be treated as one of the factors predisposing to pain. Therefore, it is worth assessing neurodynamic functions during examinations to implement appropriate physiotherapeutic interventions and assess their effectiveness. Moreover, it provides valuable information in terms of clinical practice and the prevention of pain because non-treated tensions in one area of the body may, with time, generate broader dysfunctions of the nervous system.

The present study had certain limitations. Therapists were not blind to the existence of scoliosis. Blinding the researchers was difficult due to the symptoms of scoliosis visible during the body posture examination. A significant limitation was the lack of assessment of the reliability of the ULNT1, SLR, and SLUMP tests in a group of adolescents with IS. Our observations and conclusions in this study regarding the above-mentioned tests were based on previous studies. In the future, the reliability of neurodynamic tests should be verified in the population of children and youth with IS and healthy peers. Further, the age range of the participants was wide. The intense changes occurring at the age of 10–15 years during puberty and intensive growth could have impacted the values of the assessed parameters.

Incomplete analysis of pain among the participants was another limitation of the study. The participants only confirmed or denied that they had experienced pain in the last 3 months. However, the character and intensity of pain or provoking factors were not analysed, and the validated tool (questionnaire or scale) for pain assessment was not applied. In the future, the analysis should be carried out in a larger group, taking into account various types of spine deformity.

## 5. Conclusions

Our findings revealed increased MNS in adolescents with IS. Values of neurodynamic tests correlated with the sagittal profile of the spine and mobility of the spine and lower limb joints. Therefore, the examination of adolescents with IS should include an assessment of MNS. Future studies should extend the assessment of neurodynamic functions and explore this issue to improve the understanding of the mechanisms of IS.

## Figures and Tables

**Figure 1 jcm-11-01115-f001:**
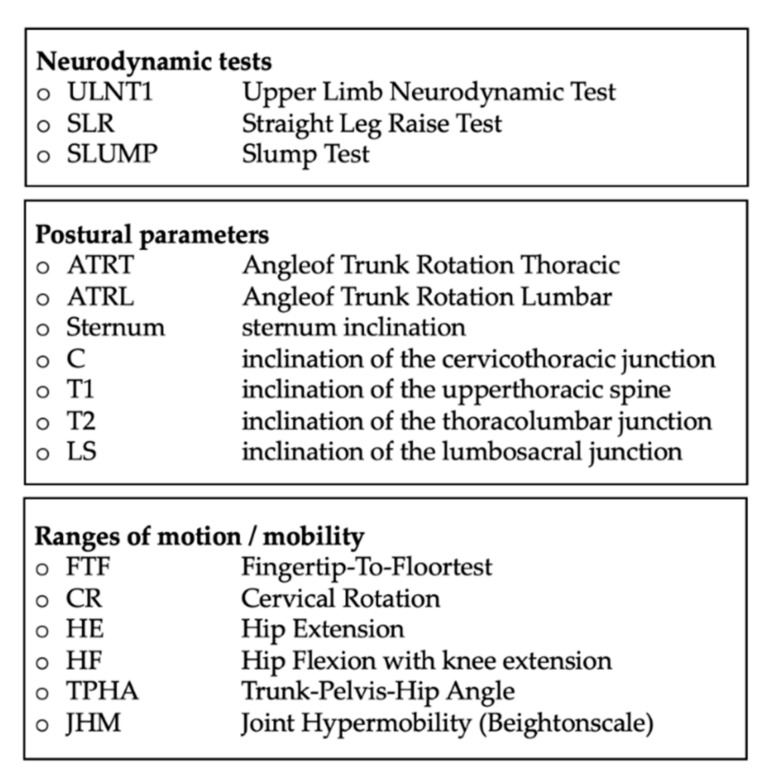
List of abbreviations for parameters used in this paper.

**Figure 2 jcm-11-01115-f002:**
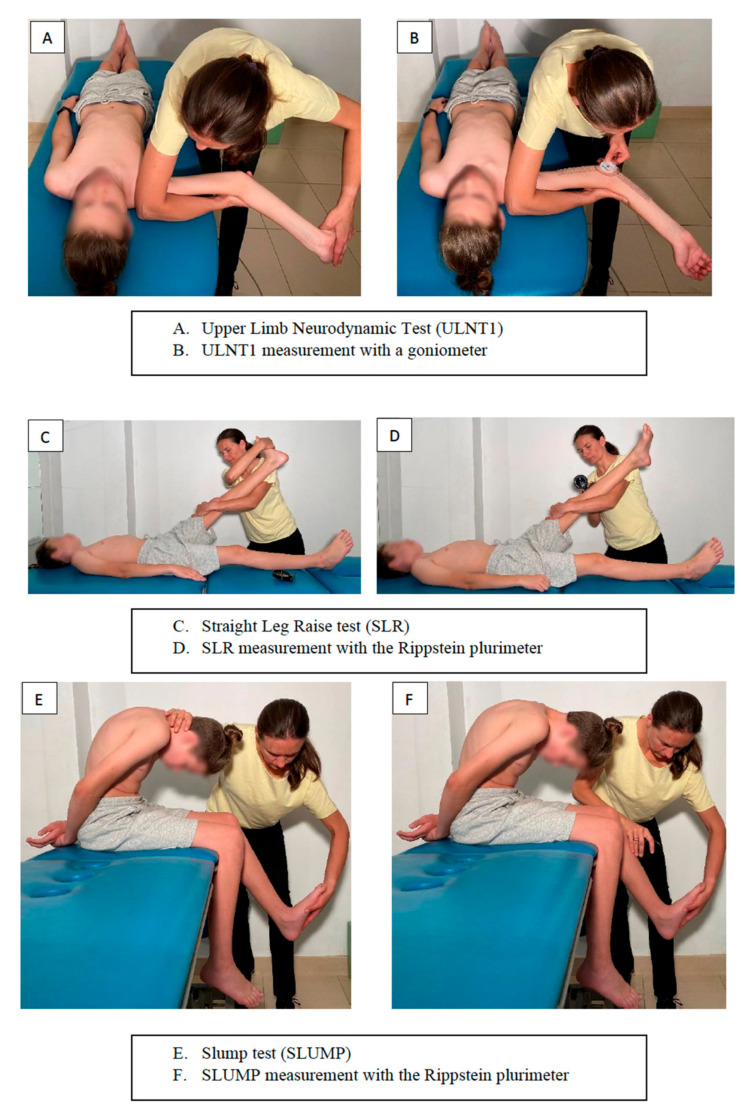
Neurodynamic tests used in this study: (**A**,**B**) ULNT1, (**C**,**D**) SLR, (**E**,**F**) SLUMP.

**Figure 3 jcm-11-01115-f003:**
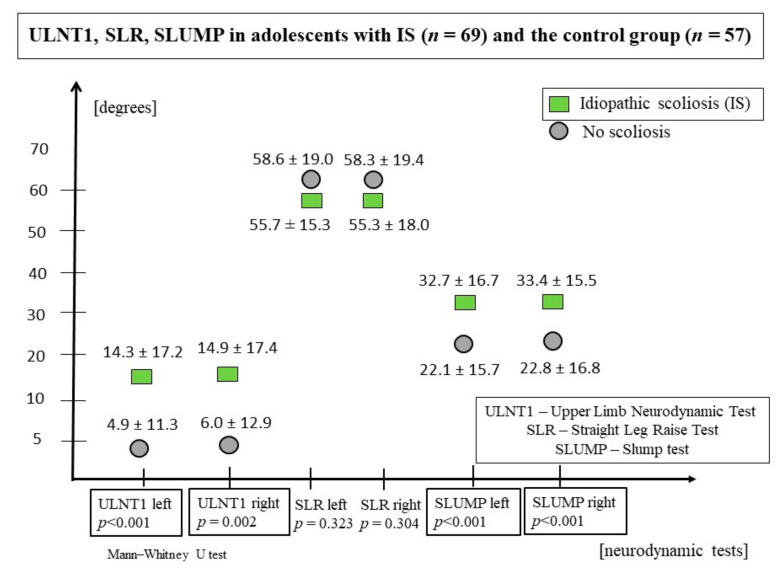
ULNT1, SLR and SLUMP values in adolescents with IS and in the control group.

**Figure 4 jcm-11-01115-f004:**
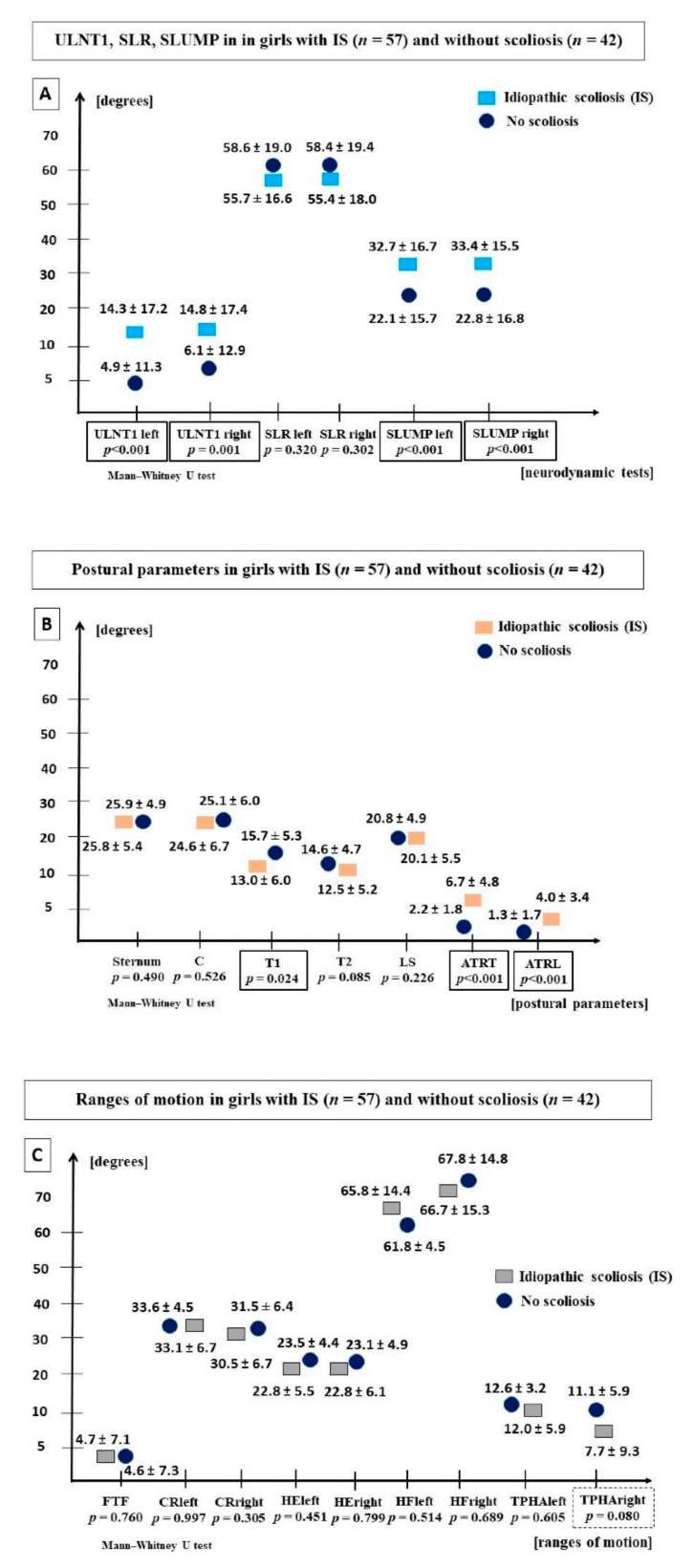
(**A**) ULNT1, SLR and SLUMP; (**B**) postural parameters; (**C**) ranges of motion in girls with IS (*n* = 57) and asymptomatic girls (*n* = 42). Abbreviations: ULNT1, Upper Limb Neurodynamic Test; SLR, Straight Leg Raise Test; SLUMP, Slump Test; ATRT, angle of trunk rotation in the thoracic spine; ATRL, angle of trunk rotation in the lumbar spine; Sternum, sternum inclination; C, inclination of the cervicothoracic junction; T1, inclination of the upper thoracic spine; T2, inclination of the thoracic-lumbar junction; LS, inclination of the lumbosacral junction; FTF, Fingertip-To-Floor test; CR, Cervical Rotation; HE, Hip Extension; HF, Hip Flexion with knee extension; TPHA, Trunk-Pelvis-Hip Angle; JHM, Joint Hypermobility; *p*, statistical significance (Mann–Whitney U test).

**Table 1 jcm-11-01115-t001:** Characteristics of adolescents with idiopathic scoliosis (IS) and the control group (C).

Characteristics	Idiopathic Scoliosis (*n* = 69)	Control Group (*n* = 57)	*p*
Age [years]			0.410
Median ± Q (range)	13.0 ± 1.5 (11.0–14.0)	12.0 ± 1.5 (11.0–14.0)	
Mean ± SD (range)	12.7 ± 1.7 (10.0–15.0)	12.4 ± 1.7 (10.0–15.0)	
Body mass [kg]			0.944
Median ± Q (range)	49.0 ± 5.0 (44.0–54.0)	48.0 ± 7.5 (40.0–55.0)	
Mean ± SD (range)	48.0 ± 10.2 (28.0–71.0)	47.9 ± 12.0 (29.0–75.0)	
Body height [cm]			0.815
Median ± Q (range)	160.0 ± 6.5 (154.0–167.0)	162.0 ± 10.0 (148.0–168.0)	
Mean ± SD (range)	159.3 ± 10.9 (132.0–181.0)	158.8 ± 12.5 (136.0 –186.0)	
Gender *n* (%)			
Girls	57 (82.6%)	42 (73.7%)	
Boys	12 (17.4%)	15 (26.3%)	
Cobb angle [°]			
Double scoliosis Th	*n* = 54 (78.3%)		
Median ± Q (range)	22.5 ± 6.5 (20.0–33.0)		
Mean ± SD (range)	26.1 ± 12.6 (11.0–69.0)		
Double scoliosis L	*n* = 54 (78.3%)		
Median ± Q (range)	23.00 ± 7.50 (15.0–30.0)		
Mean ± SD (range)	24.8 ± 11.5 (11.0–68.0)		
Single scoliosis ThL	*n* = 15 (21.7%)		
Median ± Q (range)	16.0 ± 5.0 (15.0–25.0)		
Mean ± SD (range)	24.9 ± 11.5 (11.0–45.0)		
Physiotherapy	54 (78.3%)	18 (31.6%)	<0.001
(last year) *n* (%)			
Brace *n* (%)	25 (36.2%)	-	
Pain (last 3 months) *n* (%)	26 (37.7%)	7 (12.3%)	0.001
Head			
Back pain	3 (4.3%)	0 (0.0%)	
Cervical spine	20 (29.0%)	6 (10.5%)	
Thoracic spine	5 (7.2%)	0 (0.0%)	
Lumbar spine	12 (17.4%)	4 (7.0%)	
Upper limbs	15 (21.7%)	4 (7.0%)	
Lower limbs	1 (1.4%)	0 (0.0%)	
Several body parts	4 (5.8%)	2 (3.5%)	
	13 (18.8%)	3 (5.3%)	

Abbreviations: Q, quartiles; SD, standard deviation; *n*, number of participants; *p*, statistical significance.

**Table 2 jcm-11-01115-t002:** The values of neurodynamic tests, postural parameters and ranges of motion in adolescents with idiopathic scoliosis (IS) and the control group (C).

Measurements	Idiopathic Scoliosis (*n* = 69)		Control Group (*n* = 57)		*p*
	Median ± Q	Mean ± SD	Median ± Q	Mean ± SD	
Neurodynamic tests (°)					
ULNT1left	5.0 ± 15.0 ***	14.3 ± 17.2	0.0 ± 0.0	4.9 ± 11.3	<0.001
ULNT1right	5.0 ± 15.0 **	14.9 ± 17.4	0.0 ± 0.0	6.0 ± 12.9	0.002
SLRleft	58.0 ± 15.0	55.7 ± 15.2	60.0 ± 17.5	58.6 ± 19.0	0.323
SLRright	58.0 ± 15.0	55.3 ± 18.00	60.0 ± 17.0	58.3 ± 19.4	0.304
SLUMPleft	35.0 ± 22.0 ***	32.7 ± 16.7	18.0 ± 12.0	22.1 ± 15.7	<0.001
SLUMPright	36.0 ± 21.5 ***	33.4 ± 15.5	17.0 ± 11.3	22.8 ± 16.8	<0.001
Postural parameters (°)					
ATRT	6.0 ± 3.5 ***	6.9 ± 4.9	2.0 ± 1.5	2.0 ± 1.9	<0.001
ATRL	4.0 ± 2.0 ***	4.2 ± 3.8	1.0 ± 1.0	1.4 ± 1.7	<0.001
Sternum	24.0 ± 2.5	25.3 ± 5.6	27.0 ± 3.5	26.7 ± 5.6	0.064
C	24.0 ± 5.0	24.7 ± 6.3	24.0 ± 3.0	25.7 ± 6.0	0.284
T1	13.0 ± 3.5 **	13.2 ± 6.1	16.0 ± 3.0	16.5 ± 5.0	0.001
T2	12.0 ± 4.0 **	12.3 ± 5.5	14.0 ± 3.0	14.8 ± 4.7	0.009
LS	20.0 ± 3.5	20.2 ± 5.2	20.0 ± 4.0	19.7 ± 5.8	0.331
Ranges of motion					
FTF (cm)	0.0 ± 6.5	6.5 ± 8.1	0.0 ± 5.5	5.9 ± 7.3	0.735
CRleft (°)	34.0 ± 4.0	32.9 ± 7.1	34.0 ± 3.0	32.7 ± 4.9	0.412
CRright (°)	30.0 ± 3.5	30.2 ±7.3	31.0 ± 3.0	30.1 ± 5.6	0.549
HEleft (°)	22.0 ± 2.0	23.4 ± 4.2	22.0 ± 2.0	22.05 ± 5.7	0.142
HEright (°)	22.0 ± 3.0	23.0 ± 5.6	22.0 ± 2.0	22.2 ± 5.6	0.205
HFleft (°)	68.0 ± 11.0	64.1 ± 14.8	66.0 ± 12.0	63.7 ± 16.2	0.943
HFright (°)	70.0 ± 11.0	65.0 ± 15.2	66.0 ± 12.0	63.8 ± 16.6	0.737
TPHAleft (°)	12.0 ± 2.0	11.9 ± 6.0	12.0 ± 2.0	12.1 ± 3.6	0.941
TPHAright (°)	10.0 ± 5.0	8.3 ± 9.4	10.0 ± 3.0	10.7 ± 5.8	0.165
JHM score	3.5 ± 1.5 **	3.5 ± 2.3	2.0 ± 1.5	2.3 ± 2.4	0.001

Abbreviations: ULNT1, upper limb neurodynamic test; SLR, straight leg raise test; SLUMP, slump test; ATRT, angle of trunk rotation in the thoracic spine; ATRL, angle of trunk rotation in the lumbar spine; Sternum, sternum inclination; C, inclination of the cervicothoracic junction; T1, inclination of the upper thoracic spine; T2, inclination of the thoracic-lumbar junction; LS, inclination of the lumbosacral junction; FTF, fingertip-to-floor test; CR, cervical rotation; HE, hip extension; HF, hip flexion with knee extension; TPHA, trunk-pelvis-hip angle; JHM, joint hypermobility; *n*, number of participants; *p*, statistical significance. Notes: Mann–Whitney U test; Significance of differences between IS and control group: (**)—at the level of 0.01 ≥ *p* ≥ 0.001; (***)—at the level of *p* < 0.001.

**Table 3 jcm-11-01115-t003:** Correlations between neurodynamic tests, Cobb angle, age, body mass and height in adolescents with idiopathic scoliosis (IS) and the control group (C).

**Idiopathic Scoliosis (*n* = 69)**						
	**ULNT1 Left**	**ULNT1 Right**	**SLR Left**	**SLR Right**	**SLUMP Left**	**SLUMP Right**
ULNT1 right	0.665 ***	–	–	–	–	–
SLR left	−0.518 ***	−0.497 ***	–	–	–	–
SLR right	−0.506 ***	−0.529 ***	0.950 ***	–	–	–
SLUMP left	0.313 **	0.432 **	−0.575 ***	−0.524 ***	–	–
SLUMP right	0.383 **	0.462 **	−0.545 ***	−0.514 ***	0.900 ***	–
Cobb angle	0.016	−0.073	0.197	0.177	−0.176	−0.126
Age	0.370 **	0.215	−0.113	−0.112	−0.014	0.090
Body mass	0.226	0.115	−0.186	−0.176	0.030	0.102
Body height	0.325 **	0.217	−0.212	−0.184	0.118	0.189
**Control Group (*n* = 57)**						
	**ULNT1 Left**	**ULNT1 Right**	**SLR Left**	**SLR Right**	**SLUMP Left**	**SLUMP Right**
ULNT1 right	0.928 ***	–	–	–	–	–
SLR left	−0.589 ***	−0.625 ***	–	–	–	–
SLR right	−0.601 ***	−0.622 ***	0.979 ***	–	–	–
SLUMP left	0.384 **	0.423 **	−0.650 ***	−0.644 ***	–	–
SLUMP right	0.373 **	0.426 **	−0.636 ***	−0.624 ***	0.976 ***	–
Age	0.140	0.147	−0.239	−0.218	0.190	0.184
Body mass	0.199	0.215	−0.321 *	−0.300 *	0.045	0.040
Body height	0.216	0.260	−0.316 *	−0.298 *	0.137	0.148

Abbreviations: ULNT1, upper limb neurodynamic test; SLR, straight leg raise test; SLUMP, slump test; *n*, number of participants; *p*, statistical significance. Notes: Significance of differences: (*)—at the level of 0.05 > *p* > 0.01; (**)—at the level of 0.01 ≥ *p* ≥ 0.001; (***)—at the level of *p* < 0.001.

**Table 4 jcm-11-01115-t004:** Correlations between neurodynamic tests and postural parameters in adolescents with idiopathic scoliosis (IS) and the control group (C).

**Idiopathic Scoliosis (*n* = 69)**						
	**ULNT1 Left**	**ULNT1 Right**	**SLR Left**	**SLR Right**	**SLUMP Left**	**SLUMP Right**
Sternum	0.067	−0.043	−0.010	−0.074	0.007	−0.002
C	0.425 ***	0.324 **	−0.377 **	−0.407 **	0.103	0.131
T1	0.487 ***	0.370 **	−0.450 ***	−0.432 ***	0.158	0.216
T2	0.294 *	0.175	−0.264 *	−0.281 *	0.118	0.164
LS	0.300 *	0.412 **	−0.135	−0.098	0.256 *	0.243 *
ATRT	−0.249 *	−0.282 *	0.160	0.169	−0.119	−0.198
ATRL	0.283 *	0.104	−0.124	−0.122	0.122	0.130
**Control Group (*n* = 57)**						
	**ULNT1 Left**	**ULNT1 Right**	**SLR Left**	**SLR Right**	**SLUMP Left**	**SLUMP Right**
Sternum	0.023	−0.007	−0.122	−0.104	0.045	0.059
C	0.297 *	0.356 **	−0.526 ***	−0.544 ***	0.152	0.171
T1	0.352 **	0.265 *	−0.370 **	−0.383 **	0.173	0.125
T2	0.240	0.270 *	−0.253	−0.282 *	0.191	0.200
LS	0.190	0.272 *	−0.374 **	−0.346 **	0.292 *	0.301 *
ATRT	−0.429 **	−0.419 **	0.301 *	0.297 *	−0.021	−0.066
ATRL	−0.057	−0.109	0.141	0.174	−0.109	−0.136

Abbreviations: ULNT1, upper limb neurodynamic test; SLR, straight leg raise test; SLUMP, slump test; ATR, angle of trunk rotation; T, thoracic; L, lumbar; Sternum, sternum inclination; C, inclination of the cervicothoracic junction; T1, inclination of the upper thoracic spine; T2, inclination of the thoracic-lumbar junction; LS, inclination of the lumbosacral junction; *n*, number of participants; *p*, statistical significance. Notes: Significance of differences: (*)—at the level of 0.05 > *p* > 0.01; (**)—at the level of 0.01 ≥ *p* ≥ 0.001; (***)—at the level of *p* < 0.001.

**Table 5 jcm-11-01115-t005:** Correlations between neurodynamic tests and ranges of motion in adolescents with idiopathic scoliosis (IS) and the control group (C).

**Idiopathic Scoliosis (*n* = 69)**						
	**ULNT1 Left**	**ULNT1 Right**	**SLR Left**	**SLR Right**	**SLUMP Left**	**SLUMP Right**
FTF	−0.026	−0.141	−0.396 **	−0.370 **	0.189	0.201
CR left	−0.139	−0.015	0.096	0.058	−0.105	−0.125
CR right	−0.058	0.031	0.022	−0.020	−0.178	−0.168
HE left	0.120	0.049	0.268 *	0.211	−0.159	−0.120
HE right	0.213	−0.011	0.133	0.067	−0.124	−0.095
HF left	−0.447 **	−0.426 ***	0.941 ***	0.905 ***	−0.533 ***	−0.485 ***
HF right	−0.403 **	−0.347 **	0.895 ***	0.886 ***	−0.465 ***	−0.432 ***
TPHA left	0.206	0.202	−0.078	−0.101	−0.011	0.029
TPHA right	0.364 **	0.398 **	−0.350 **	−0.347 **	0.124	0.151
JHM	−0.136	−0.099	0.123	0.108	−0.082	−0.077
**Control Group (*n* = 57)**						
	**ULNT1 Left**	**ULNT1 Right**	**SLR Left**	**SLR Right**	**SLUMP Left**	**SLUMP Right**
FTF	0.214	0.245	−0.543 ***	−0.504 ***	0.433 **	0.462 ***
CR left	0.025	−0.009	0.188	0.199	−0.275 *	−0.254
CR right	−0.097	−0.172	0.273 *	0.267 *	−0.477 ***	−0.462 ***
HE left	−0.045	−0.096	0.228	0.207	−0.026	−0.066
HE right	−0.192	−0.196	0.281 *	0.279 *	0.005	−0.030
HF left	−0.544 ***	−0.572 ***	0.965 ***	0.939 ***	−0.604 ***	−0.589 ***
HF right	−0.561 ***	−0.581 ***	0.952 ***	0.955 ***	−0.586 ***	−0.571 ***
TPHA left	0.350 **	0.353 **	−0.099	−0.113	0.208	0.215
TPHA right	0.303 *	0.250	0.180	−0.205	0.173	0.153
JHM	0.095	0.027	0.077	0.047	−0.017	−0.054

Abbreviations: ULNT1, upper limb neurodynamic test; SLR, straight leg raise test; SLUMP, slump test; ATR, angle of trunk rotation; Sternum, sternum inclination; C, inclination of the cervicothoracic junction; T1, inclination of the upper thoracic spine; T2, inclination of the thoracic-lumbar junction; LS, inclination of the lumbosacral junction; FTF, fingertip-to-floor test; CR, cervical rotation; HE, hip extension; HF, hip flexion with knee extension; TPHA, trunk-pelvis-hip angle; JHM, joint hypermobility; *n*, number of participants; *p*, statistical significance. Notes: Mann–Whitney U test; significance of differences between IS and control group: (*)—at the level of 0.05 > *p* > 0.01; (**)—at the level of 0.01 ≥ *p* ≥ 0.001; (***)—at the level of *p* < 0.001.

## Data Availability

The data presented in this study are available on request from the corresponding author.

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
