# Peer review of "Neurodynamic Functions and Their Correlations with Postural Parameters in Adolescents with Idiopathic Scoliosis"

_jcm, 2022, doi:10.3390/jcm11041115_

Round 1

Reviewer 1 Report

  1. General remarks:

- the authors made a comparative study between two lots of adolescents/ children (?), one with adolescents having idiopathic scolisis an a control one, with healthy subjects

- in this purpose, the authors have made, in both lots, quantified evaluations using three (considered in the literature) neurodynamic (clinical-functional assessment) tests, and respectively, also - to make possible the related designed comparisons - seven postural parameters and respectively, seven range of motion/ mobility measurement items/ parameters, plus (general morphometric data: body height and mass), in each of the enrolled participants

- it is an endeavor that entailed a commeandably consistent amount of work

  1. Specific remarks and suggestions:

- the definition of the PNS physiologic roles is quite incomplete: ”The function of the peripheral nervous system (PNS) is to carry sensory information” ? (and this assertion is repeated), because the PNS is involved, as well, in carrying motor and vegetative, afferents and efferents

- adolescence has to be briefly but clearly defined and specified, as age of onset and age until it is considered to end, including in this manuscript – bibliographic suggestion: Thakur R.; Gautam R., Differential onset of Puberty and Adolescence among girls and boys of a Central Indian Town (Sagar), The Oriental Anthropologist, 2017, 17, 137-147.

- the authors chose to make a comparative ”assessment of neurodynamic functions” between two lots, i. e. ”69 adolescents (?) with IS and 57 healthy children (?), aged 10-15 years” (?): the age interval of 10-15 years practically overlaps the end of childhood, and the onset of puberty, respectively – over ~ 13 years old – the onset of adolescence (see the above mentioned/ indicated bibliographic resource)

- the enrolled/ participant adolescents or their ”Legal guardians of children gave a written consent to participate in the study”, but did they give their accept that their images (including with the face) to appear published in this article ?

-  regarding the statistical analysis:

  1. a) an explanation for the lack of normality/ homogeneity of the populations in both lots and in all variables would be appropriate, if possible
  2. b) in Tables 3, 4 and 5 there are included a lot of correlation coefficients, plus the afferent p values. Because such a p value represents the probability of that correlation coefficient to be 0, i. e. to reflect the statistical independence between the respective two variables, many of these p's are quite useless and unnecessary complicate the respective tables. So, a non-mandatory proposal/ recommendation would be to simplify the tables (although the authors have strictly applied the related taxonomy presented in the bibliographic reference they have quoted: ”Mukaka, M.M. Statistics corner: a guide to appropriate use of correlation coefficient in medical research. Malawi Med J. 2012, 24:69–71.”), and emphasize: 1) the ”big” correlation coefficients, i. e. close to 1 or -1 (those indicating towards a linear dependence between the respective variables), and 2) the ”small” correlation coefficients (which indicate towards the statistical independence of the respective variables); the other p values are not so important. Instead, making thus an editorial space economy, it would be appropriate that the authors to complete their Spearman's rho computing with a respective Pearson's r correlation, and then with a ”multidimensional statistical analysis”, and even, maybe, also with a ”principal components” one, in order to detect the possible dependence of a variable of several others.

- The references are not very new: less than half of them are within the 10 years, and of these, less than ten, are within the last 5 years

Author Response

Dear Reviewer 1

Thank you very much for your time devoted to analyzing our manuscript and valuable advice. Thank you very much for your questions, suggestions, and support. We have revised the manuscript as suggested. Below are the answers to individual questions.

Comments and Suggestions for Authors

  1. General remarks:

- the authors made a comparative study between two lots of adolescents/ children (?), one with adolescents having idiopathic scoliosis and a control one, with healthy subjects

Reply: We have included the answer regarding adolescents below.

- in this purpose, the authors have made, in both lots, quantified evaluations using three (considered in the literature) neurodynamic (clinical-functional assessment) tests, and respectively, also - to make possible the related designed comparisons - seven postural parameters and respectively, seven range of motion/ mobility measurement items/ parameters, plus (general morphometric data: body height and mass), in each of the enrolled participants

- it is an endeavor that entailed a commeandably consistent amount of work

  1. Specific remarks and suggestions:

- the definition of the PNS physiologic roles is quite incomplete: ”The function of the peripheral nervous system (PNS) is to carry sensory information” ? (and this assertion is repeated), because the PNS is involved, as well, in carrying motor and vegetative, afferents and efferents

Reply: Thank you, indeed this sentence was not complete. At the beginning of the Introduction, we completed the information on the peripheral nervous system.

- adolescence has to be briefly but clearly defined and specified, as age of onset and age until it is considered to end, including in this manuscript – bibliographic suggestion: Thakur R.; Gautam R., Differential onset of Puberty and Adolescence among girls and boys of a Central Indian Town (Sagar), The Oriental Anthropologist, 2017, 17, 137-147.

Reply: Thanks for your suggestion. You are indeed right that the age of the participants and maturation could have a significant impact on the results of the research. In the literature, however, we have not found an official division taking into account the onset of puberty. When planning the selection of the group, we were guided by the division of WHO. WHO defines an age for adolescence between 10-18 years of age, and some authors suggest extending this period. In publications on the prevalence of back pain in adolescents with scoliosis, adolescents aged 10-18 were also qualified for the study (Wong, 2019; Teles, 2020). However, bearing in mind that puberty affects many body functions, including postural parameters, we commented on this issue in the discussion and cited the article you mentioned (Thakur R.; Gautam R., Differential onset of Puberty and Adolescence among girls and boys of a Central Indian Town (Sagar), The Oriental Anthropologist, 2017, 17, 137-147). Additionally, in Table 3, we added the correlations between the values ​​of neurodynamic tests and age.

- the authors chose to make a comparative ”assessment of neurodynamic functions” between two lots, i. e. ”69 adolescents (?)with IS and 57 healthy children (?), aged 10-15 years” (?): the age interval of 10-15 years practically overlaps the end of childhood, and the onset of puberty, respectively – over ~ 13 years old – the onset of adolescence (see the above mentioned/ indicated bibliographic resource)

Reply: We corrected this sentence. We used the word "adolescents" as defined by the WHO, but we mentioned changes in adolescence in the discussion. We think that it is difficult to really precisely define the beginning of "adolescents". We provide therapy for girls with postural disorders who begin puberty sometimes as early as 10-11 years of age, and sometimes after 13 years of age, and in boys, the beginning of puberty usually takes place later. Therefore, in our opinion, it is difficult to plan studies in larger groups strictly taking into account the onset of puberty.

- the enrolled/ participant adolescents or their ”Legal guardians of children gave a written consent to participate in the study”, but did they give their accept that their images (including with the face) to appear published in this article?

Reply: Of course, you are right that we should think about it. The photo shows one of the authors of the article with her son. She has agreed to share these photos. However, bearing in mind your doubts, we have hidden the boy's face in Figure 2.

-  regarding the statistical analysis:

  1. a) an explanation for the lack of normality/ homogeneity of the populations in both lots and in all variables would be appropriate, if possible.
  2. b) Tables 3, 4, and 5 there are included a lot of correlation coefficients, plus the afferent p values. Because such a p-value represents the probability of that correlation coefficient to be 0, i. e. to reflect the statistical independence between the respective two variables, many of these p's are quite useless and unnecessarily complicate the respective tables. So, a non-mandatory proposal/ recommendation would be to simplify the tables (although the authors have strictly applied the related taxonomy presented in the bibliographic reference they have quoted: ”Mukaka, M.M. Statistics corner: a guide to the appropriate use of correlation coefficient in medical research. Malawi Med J. 2012, 24:69–71.”), and emphasize 1) the ”big” correlation coefficients, i. e. close to 1 or -1 (those indicating towards a linear dependence between the respective variables), and 2) the ”small” correlation coefficients (which indicate towards the statistical independence of the respective variables); the other p values are not so important. Instead, making thus an editorial space economy, it would be appropriate that the authors complete their Spearman's rho computing with a respective Pearson's r correlation, and then with a ”multidimensional statistical analysis”, and even, maybe, also with a ”principal components” one, in order to detect the possible dependence of a variable of several others.

Reply:

  1. In the Methods section, we have completed the information on a normal distribution.
  2. We discussed the analysis with a professional statistician and made additional calculations to complete the data presented in the article. In connection with the use of non-parametric tests in Table 1 and Table 2, we added the values ​​of Median ± Q. However, we left the values ​​of Mean ± SD because we think that these values ​​will be useful for practitioners dealing with youth therapy. In Table 3, Table 4, and Table 5, we removed the p-values ​​and kept the values ​​of the correlation coefficient. After consulting with the statistician, we did not delete the reference to Mukak's article. The value of p shows whether the change is statistically significant, and the value of the coefficient indicates its strength. Therefore, it seems to us that these two methods of assessment are not contradictory.

- The references are not very new: less than half of them are within the 10 years, and of these, less than ten, are within the last 5 years

Reply: We added new literature and dropped some older publications. Currently, the list of publications includes 15 articles published in the last 5 years. However, we did not delete some of the older important articles.

We kindly thank Reviewer 1 for her/his positive feedback and insightful comments. We ensure that the entire manuscript has been thoroughly revised. All the Reviewer's comments have been considered and have contributed to improving the quality of our paper, which we would like to thank you for.

We very much hope that our carefully prepared, point-by-point reply, appears comprehensive and proves helpful in obtaining a positive final decision accepting our paper for publication in your prestigious Journal of Clinical Medicine.

Sincerely,

The authors.

Reviewer 2 Report

Abstract:

The main reason for scoliosis treatment is prevention of progression, restoration of truncal balance and avoidance of (cardio-) pulmonary problems, not treatment of pain. Of course pain reduction can be an element of conservatieve scoliosis treatment but it is usually not the aim of surgical treatment.

Introduction:

- 29. The function of the peripheral nervous system (PNS) is to carry sensory information. I believe there are more functions to the PNS.

- 30. It has three major mechanical functions, i.e. contraction, sliding and compression. I do not understand what is meant here.

- 30. Disorders in these functions which lead to nerve irritation or inflammation increase nerve sensitivity to mechanical stimuli and may cause pain, although the basic function of the PNS, i.e. carrying sensory information, is maintained [3,4]. The cited literature has a different meaning than what is stated here.

- 55. Studies revealed that pain occurred in individuals with IS both during conservative treatment and after the surgery [22-24]. The cited literature is about patients many years after brace or surgical treatment, not (as is stated) during.

-74. The criteria for inclusion to the study group were following: girls and boys aged 1015 years, This is a wide age range, with undoubtedly very different stages of maturation.

-81. Methods. I assume the therapists were not blinded to the existence of scoliosis, at least it is not mentioned in the text.

-90. The type of pain, frequency of occurrence and level of severity were not assessed. Many scoliosis patients are known to have mild to moderate pain, especially during periods of rapid growth. It would have helped to have at least a VAS-score and information on whether it occurred during day or night. Also, it does not become clear if the pain was experienced in the spine, trunc, extremities or perhaps the head?

-103. In the provided table with Parameter abbreviations, Cobb angle is mentioned under Postural parameters. I don’t believe it is one.

-107 and further. The described tests are presented as signs of dysfunctioning of the Peripheral Nervous System. It would be helpful to have a reference for each of the tests, its reproducibility and reliability in this part of the manuscript.

-184. Within this group, the Cobb angle range was 11° to 69°. Although both can be called scoliosis by definition, this is a large spread of severeness of the curve.

-190. In Table 1 the values for Cobb angle are missing.

-229. Values of ATR in the thoracic and lumbar segment of the spine as well as Beighton score values were significantly higher in individuals with scoliosis. Measurements with the plurimeter revealed decreased values of T1 and T2 in the IS group compared to the control group, which indicates decreased kyphosis in patients with scoliosis. No significant differences were noted between the values of the measurements at the sternum, cervical-thoracic (C) and lumbosacral junction (LS). Ranges of movement in the cervical spine, in hip joints and in the FTF test did not differ significantly either (Table 2). These observations appear coorespond to what is known of patients with scoliosis, perhaps except for the finding of related to joint mobility.

-246. No significant relationship was found between the values of neurodynamic tests and the values of the Cobb angle in adolescents with IS. Given the rather wide range of Cobb angles, this is surprising.

-264. The analysis showed several relationships between neurodynamic tests and ranges of motion. In both groups, a significant correlation was found between the ULNT1, SLR and SLUMP tests, and the range of flexion in the hip joint with the knee extended (HF). Is this not what can be expected given the similarities between what is tested, at least for the tests concerning the lower extremities?

-322.  growth spur = growth spurt

-324. One of the causes of pain may be impairments of neurodynamic functions. Therefore it would have been very helpful to describe in more detail the character of the experienced pain, see comment about -90. The pain that is sought for in this study seems to be more radiating pain than axial pain that is often described in scoliosis patients.

-328. For this study, three neurodynamic tests whose reliability has been confirmed in different groups of participants were selected, i.e. ULNT [13,14,15], SLR [11,16,17,20] and SLUMP test [18,19]. Referencve 20 is about patients with cerebral palsy.

-332. The results showed that the tension of PNS in adolescents with IS assessed with the use of ULNT1 and SLUMP was larger than in the control group. Given the fact that the investigators were not blinded to the existence of a scoliosis might constitute a bias. Also, these findings could be influenced by different stages of maturation in the examined age range of 10-15.

-343. Also, no norms for neurodynamic functions have been defined in the groups of healthy children and youth. This is an important statement.

-359. This phenomenon requires further analysis depending on the type of scoliosis, the location of the apex of the curvature and other factors in a larger group of adolescents with IS. And different maturation stages.

Author Response

Dear Reviewer 2

Thank you very much for your time devoted to analyzing our manuscript and valuable advice. Thank you very much for your questions, suggestions and support. We have revised the manuscript as suggested. Below are the answers to individual questions.

Abstract:

The main reason for scoliosis treatment is prevention of progression, restoration of truncal balance and avoidance of (cardio-) pulmonary problems, not treatment of pain. Of course pain reduction can be an element of conservatieve scoliosis treatment but it is usually not the aim of surgical treatment.

Reply: You are right. Pain reduction is only one of goals and not the most important one. In this text, we refer to the various goals of treating patients with scoliosis.

Introduction:

- 29. The function of the peripheral nervous system (PNS) is to carry sensory information. I believe there are more functions to the PNS.

Reply. We supplemented the text in the introduction.

- 30. It has three major mechanical functions, i.e. contraction, sliding and compression. I do not understand what is meant here.

Reply: We agree that the text was not fully understandable and the introduction was not completed. These concepts are derived from manual therapy and neurodynamic. We improved the text and referred to the neurodynamic literature. We hope the text will now be clearer.

- 30. Disorders in these functions which lead to nerve irritation or inflammation increase nerve sensitivity to mechanical stimuli and may cause pain, although the basic function of the PNS, i.e. carrying sensory information, is maintained [3,4]. The cited literature has a different meaning than what is stated here.

Reply: Thank you for your opinion, these articles were not strictly related to the topic of this research. In the introduction, we used other articles that better fit the topic of the study.

- 55. Studies revealed that pain occurred in individuals with IS both during conservative treatment and after the surgery [22-24]. The cited literature is about patients many years after brace or surgical treatment, not (as is stated) during.

Reply: We removed articles 22 and 23 on the symptoms in patients many years after brace and surgery treatment. We've added some newer articles on pain in adolescents with scoliosis. Thanks for your feedback. These newer publications brought much valuable information and gave the possibility to develop the discussion.

-74. The criteria for inclusion to the study group were following: girls and boys aged 1015 years, This is a wide age range, with undoubtedly very different stages of maturation.

Reply: We have included your comments in the manuscript. Indeed, there are different stages of maturation during this period. Physiological changes during this period may affect posture and body mobility. We have included a wide age range as a research limitation. Additionally, we made a correlation between age and the values ​​of neurodynamic tests (Table 3).

-81. Methods. I assume the therapists were not blinded to the existence of scoliosis, at least it is not mentioned in the text.

Reply: Therapists were not blinded. We have added an explanation in the Methods section and in the study limitations.

-90. The type of pain, frequency of occurrence and level of severity were not assessed. Many scoliosis patients are known to have mild to moderate pain, especially during periods of rapid growth. It would have helped to have at least a VAS-score and information on whether it occurred during day or night. Also, it does not become clear if the pain was experienced in the spine, trunc, extremities or perhaps the head?

Reply: Unfortunately, we did not use a score scale. On the other hand, the participants were asked about the location of the pain. We have added this information in Table 2 and in the discussion.

-103. In the provided table with Parameter abbreviations, Cobb angle is mentioned under Postural parameters. I don’t believe it is one.

Reply: Thanks for your suggestion. We removed the Cobb angle from the postural parameters in Figure 1.

-107 and further. The described tests are presented as signs of dysfunctioning of the Peripheral Nervous System. It would be helpful to have a reference for each of the tests, its reproducibility and reliability in this part of the manuscript.

Reply: Information on the reliability of the tests was included in the Introduction. We have now additionally cited relevant articles in the Methods section.

-184. Within this group, the Cobb angle range was 11° to 69°. Although both can be called scoliosis by definition, this is a large spread of severeness of the curve.

Reply: Yes, we are aware of that. Therefore, we checked the correlation between the Cobb angle and the values ​​of neurodynamic tests. In addition, a comparison between groups of participants with lower and greater scoliosis (over 30 degrees) was made. It was shown that participants with more deformity showed more nervous system tension (SLUMP test).  We mentioned a large spread of severeness of scoliosis in the study group in the discussion.

-190. In Table 1 the values for Cobb angle are missing.

Reply: The Cobb values ​​were in Table 1, you may not have noticed them because they were listed below as double-curve and single-curve scoliosis. We have added information about the minimum and maximum values ​​of the Cobb angles.

-229. Values of ATR in the thoracic and lumbar segment of the spine as well as Beighton score values were significantly higher in individuals with scoliosis. Measurements with the plurimeter revealed decreased values of T1 and T2 in the IS group compared to the control group, which indicates decreased kyphosis in patients with scoliosis. No significant differences were noted between the values of the measurements at the sternum, cervical-thoracic (C) and lumbosacral junction (LS). Ranges of movement in the cervical spine, in hip joints and in the FTF test did not differ significantly either (Table 2). These observations appear coorespond to what is known of patients with scoliosis, perhaps except for the finding of related to joint mobility.

Reply: Yes, we agree that all of the above results confirm the results of other researchers. In the discussion, we referred to relevant publications, also on hypermobility.

-246. No significant relationship was found between the values of neurodynamic tests and the values of the Cobb angle in adolescents with IS. Given the rather wide range of Cobb angles, this is surprising.

Reply: You are right, this is surprising. We also expected a significant relationship, especially as the comparison between the groups with scoliosis below 30 degrees showed significant differences. This means that besides the Cobb angle, there are other factors that influence the tone of the peripheral nervous system. In the discussion, we mentioned the lack of correlation between the Cobb angle and the values ​​of the neurodynamic tests.

-264. The analysis showed several relationships between neurodynamic tests and ranges of motion. In both groups, a significant correlation was found between the ULNT1, SLR and SLUMP tests, and the range of flexion in the hip joint with the knee extended (HF). Is this not what can be expected given the similarities between what is tested, at least for the tests concerning the lower extremities?

You are right, but it wasn't analyzed in the past. Indeed, the HF and SLR tests for the lower limbs are similar, but in the case of pain in patients, the values ​​of these tests may differ significantly. On the other hand, the similarity between HF and ULNT and SUMP tests is smaller and a significant correlation has also been demonstrated. These dependencies require some thought and require evaluation.

-322.  growth spur = growth spurt

Thank you! It was corrected.

-324. One of the causes of pain may be impairments of neurodynamic functions. Therefore it would have been very helpful to describe in more detail the character of the experienced pain, see comment about -90. The pain that is sought for in this study seems to be more radiating pain than axial pain that is often described in scoliosis patients.

Reply: We added information expected in line 90. Neurodynamic tests are also useful for examining people with back pain. Patients with discopathy often report major pain centrally in the cervical, thoracic and lumbar regions. For some, the pain radiates to the extremities. It seems to us that the tests can be useful in both cases for clinical reasoning. We have added information on the location of the pain in the manuscript. Back pain was reported by the highest number of participants, which confirms the results of other researchers. We referred to these results in the discussion.

-328. For this study, three neurodynamic tests whose reliability has been confirmed in different groups of participants were selected, i.e. ULNT [13,14,15], SLR [11,16,17,20] and SLUMP test [18,19]. Referencve 20 is about patients with cerebral palsy.

Reply: Yes, we decided to include this study, because it is performed in the child population. In the manuscript, we listed different groups of patients assessed by neurodynamic tests, including children with pain and children with cerebral palsy. Unfortunately, it is difficult to find any reports of children in the literature, so each article is valuable. We emphasized in the discussion and conclusions that it is important to examine the reliability of tests and define norms in the groups of young people with IS and healthy.

-332. The results showed that the tension of PNS in adolescents with IS assessed with the use of ULNT1 and SLUMP was larger than in the control group. Given the fact that the investigators were not blinded to the existence of a scoliosis might constitute a bias. Also, these findings could be influenced by different stages of maturation in the examined age range of 10-15.

Reply: You are right. We mention this as in Method section and in discussion as a limitation of the study. The researchers could not be blinded due to the symptoms of scoliosis visible during the examination.

-343. Also, no norms for neurodynamic functions have been defined in the groups of healthy children and youth. This is an important statement.

Reply: We know our research is not perfect. We are aware that there is still a lot of research to be done in this area in the future.

-359. This phenomenon requires further analysis depending on the type of scoliosis, the location of the apex of the curvature and other factors in a larger group of adolescents with IS. And different maturation stages.

Reply: Thank you for your advice. We added this information in the discussion and conclusions.  

We kindly thank Reviewer 2 for her/his positive feedback and insightful comments. We ensure that the entire manuscript has been thoroughly revised. All the Reviewer's comments have been considered and have contributed to improving the quality of our paper, which we would like to thank you for.

We very much hope that our carefully prepared, point-by-point reply, appears comprehensive and proves helpful in obtaining a positive final decision accepting our paper for publication in your prestigious Journal of Clinical Medicine.

Sincerely,

The authors.

Reviewer 3 Report

This manuscript describes a thorough review of clinical symptoms of the peripheral nervous system among patients with adolescent idiopathic scoliosis.  Peripheral neural symptoms are indeed overshadowed by back pain and body image issues in this patient population.  It may not be surprising that the peripheral neural structures are under tension given the deformity of the spine in the AIS population.  The authors found worse PNS symptoms in patients with worse deformities as well, which makes sense.

A great deal of work went into this interesting study.  It has little clinical utility but does highlight that the AIS population has symptoms beyond those associated with the spine itself

Round 2

Reviewer 2 Report

 For instance, in the abstract the authors state: Pain prevention and reduction is one of the goals of the treatment of idiopathic scoliosis (IS), an impairment of the peripheral nervous system (PNS) that may cause pain.

The statement that AIS is an impairment of the peripheral nerve system is not one of the accepted theories of scoliosis etiology nor pathogenesis.

It feels like the study is organized around this presumption and subsequently arguments are found to support it. Pain may arise from many other causes than 'impairment of the peripheral nerve system', and scoliosis is thought not primarily to be caused by it.

Author Response

Response to Reviewer 2 Comments

Dear Reviewer,

You are right, that an impairment of the peripheral nerve system is not one of the accepted theories of scoliosis etiology nor pathogenesis. We also know that pain can be triggered by many factors.

It was not our intention to emphasize in this study that the peripheral nervous system may induce the development of scoliosis.

We tried to organize the manuscript according to your comments. We have made appropriate changes in the manuscript.

Thank you for your attention and reading our text carefully.

Point 1

For instance, in the abstract the authors state: Pain prevention and reduction is one of the goals of the treatment of idiopathic scoliosis (IS), an impairment of the peripheral nervous system (PNS) that may cause pain.

Response 1  

We agree with your comment. We did not want to focus primarily on pain,  but on assessing the neurodynamic functions in adolescents with IS. Additionally, we noticed that the first sentence of the abstract was changed during the translation by a native speaker in order to shorten the abstract and adapt it to the requirements of the journal. Sorry, we have not noticed previously that the meaning of this sentence was changed.

The text at the beginning of the abstract has been corrected to match the purpose of the study and to avoid emphasizing pain.

Point 2

The statement that AIS is an impairment of the peripheral nerve system is not one of the accepted theories of scoliosis etiology nor pathogenesis.

Response 2

We agree with your opinion. We emphasized that impairment of neurodynamic functions of the peripheral nervous system is not one of the accepted theories of scoliosis etiology /pathogenesis. We have included this information in the abstract, discussions and conclusions.

Point 3

It feels like the study is organized around this presumption and subsequently arguments are found to support it. Pain may arise from many other causes than 'impairment of the peripheral nerve system', and scoliosis is thought not primarily to be caused by it.

Response 3

Our goal was not to create a new theory and look for arguments to support it. The purpose was investigate neurodynamic functions in adolescents with scoliosis. We tried to improve the manuscript taking into account your comments.

We have changed the beginning of the abstract and introduction to refer to neurodynamic functions of the peripheral nervous system.

We limited the information on pain, because pain was not the primary goal of our study.  We have removed the text on pain from the abstract, and we have included the information in the text that the tension of the peripheral nervous system may be only one of the factors causing pain (Discussion).                           

Thank you for your all suggestions and support. We hope that our corrections improved the quality of the manuscript.

Yours sincerely

Authors